# Descriptive epidemiology of the prevalence of adolescent active travel to school in Asia: a cross-sectional study from 31 countries

Rizka Maulida [1,2] Erika Ikeda [1] Tolu Oni,[1] Esther M F van Sluijs[1]

[1]MRC Epidemiology Unit, University of Cambridge, Cambridge, UK
[2]Department of Epidemiology, Faculty of Public Health, Universitas Indonesia, Depok, Indonesia

**Correspondence to**
Rizka Maulida;
rizka.maulida@mrc-epid.cam.ac.uk

## ABSTRACT

**Objective** This study aimed to examine the prevalence of adolescent active travel to school (ATS) across 31 countries and territories in Asia, overall and by age group, sex and body mass index (BMI) category.

**Design** Cross-sectional study.

**Setting** 31 Asian countries.

**Participants** 152 368 adolescents aged 13–17 years with complete data for age, sex, measured weight and height and active travel to school from 31 Asian countries from the Global School-based student Health Survey (GSHS).

**Primary outcome** Self-reported active travel to school categorised into passive (0 days) and active (1–7 days).

**Results** Overall prevalence of adolescent ATS in Asia based on random-effect meta-analysis was 55%, ranging from 18% (UAE) to 84% (Myanmar). There was limited subregional variation: 47% in the Eastern Mediterranean (EM), 56% in the South East Asia and 64% in the Western Pacific. Summarised by random-effect meta-analysis, being an older adolescent aged 16 years and older (vs younger age below 16 years: OR: 1.08; 95% CI: 1.00 to 1.16) was positively associated with ATS. This association was strongest in EM countries. Summarised by random-effect meta-analysis, females (vs males: OR: 0.79; 95% CI: 0.71 to 0.89) and adolescents with overweight/obesity (vs underweight and normal BMI: OR: 0.92; 95% CI: 0.86 to 0.99) were less likely to use ATS. Association with sex was strongest in EM countries. Heterogeneity was considerable in all meta-analyses.

**Conclusion** The prevalence of adolescent ATS in Asia varies substantially. Overall, older and male adolescents, and adolescents with underweight and normal BMI category are more likely to actively travel to school. However, the main contributor to differences in ATS between and within regions remain unknown. Although there is substantial scope for improving ATS rates in Asia, any policy actions and interventions should be cognisant of local built, social and natural environmental contexts that may influence active travel behaviour.

## INTRODUCTION

Globally, non-communicable diseases (NCDs), notably cancers, cardiovascular diseases, type 2 diabetes and chronic respiratory illness, are responsible for about 40 million deaths each year.[1] NCDs have been associated with various

### Strengths and limitations of this study

► This study pooled comparable estimates on active travel to school from 152 368 adolescents from 31 countries in Asia.
► Data were collected using standardised sampling and data collection methods, enabling comparison across countries and subregions.
► Data were from low-income and middle-income countries; thus, the extent to which these findings are generalisable to other, particularly high-income, countries in Asia is unclear.
► Data collection was conducted in schools, and thus the conclusions drawn here are only relevant to adolescents in school.

modifiable behavioural risk factors including tobacco use, physical inactivity, unhealthy diet and the harmful use of alcohol.[1 2] The risk of developing NCDs can be lowered by avoiding these unhealthy behaviours, for example, by being physically active.[3 4] Physical inactivity-related diseases mostly manifest in adulthood but their development often starts in childhood and adolescence.[5]

Studies have reported that health benefits gained from physical activity in adolescence, defined as age 10–19 years,[6] are carried forward into adulthood.[7–9] Despite these benefits, global research on physical activity using self-reported questionnaires have shown that >80% of adolescents are not meeting international guidelines of engaging in 60 min of moderate-to-vigorous physical activity on average per day.[10] Moreover, evidence suggests that physical activity starts to decline from childhood and continues to decline into adulthood.[11–13] By contrast, a continuous high level of physical activity in general, and maintained active travel, for example, walking or cycling, throughout adolescence significantly predicts a high level of adult physical activity.[14 15] Therefore, as

adolescence is a key period for establishing active living habits, promoting active travel in this period is important to increase overall physical activity among adolescents. Given that the period of adolescence is also the age range for school attendance, active travel to school (ATS) could be an important focus for interventions to increase active travel in this age group.

Previous research in 63 low-income and middle-income countries (LMICs) showed that active travel to school is positively associated with adolescents' physical activity,[16] supporting the findings of a systematic review.[17] A single walk to school has been estimated to contribute between 16% and 18% of younger adolescents' daily moderate-to-vigorous physical activity.[18] However, data on the prevalence of adolescent ATS are scarce. A study using data from the Global Matrix 3.0 showed that half of children and adolescents actively travelled to and from places, where these trips are mainly trips to and/or from school.[19] However, this study was unable to generate a pooled estimate of the global prevalence of active travel due to different methodologies used in the countries studied. A more recent study, the 63 LMICs study, revealed a prevalence of 56.1%, but the study did not have a specific overall prevalence for Asia as well as comparisons between countries.[16] Using a mix of 27 Asian and Pacific countries, another study reported a prevalence of 42.1%.[20] Therefore, to date, no studies have solely focused their analyses on Asia, provided data by Asian subregions, nor provided data by BMI.[16 19 20]

Asia is the largest and most populous continent, with a population of 4.6 billion,[21] where many countries are emerging economies. Rapid urbanisation and dynamically changing built environments due to infrastructure development in the region can be obstacles for active travel. Additionally, these obstacles also pose a broader risk to planetary health.[22] For example, the cities with the worst air pollution levels globally are within greater Asia,[23] and air pollution has negative impacts both on the climate and health, such as respiratory and cardiovascular diseases and cancer.[24] Due to the health risks, physical activity outdoors can be unadvisable. Therefore, future interventions for active travel in this region will have to consider the specific characteristics of built environment and air pollution trends. However, interventions need to be based on an understanding of the pre-existing prevalence in this population group and region; the paucity of evidence on the prevalence of ATS across Asia poses a challenge for evidence-informed policies to promote ATS.

A few studies have been conducted in limited East and Southeast Asian countries. However, these studies had relatively small sample sizes (between 330 and 1518 participants),[25–30] were not able to compare across regions or look at differences by age or body mass index (BMI) category. Evidence from these limited Asian studies suggests that boys are more likely to use ATS compared with girls, and students living in less wealthy areas or studying in less wealthy areas are more likely to use active travel than their wealthier counterparts. ATS is also positively associated with (perceived and objectively measured) shorter distance to school, favourable built environments (eg, larger street block size and tree cover) and higher social interaction.[26–30]

The goal of this study was to examine the prevalence of adolescent ATS across 31 countries and territories in Asia, overall and by age group, sex and BMI category.

## METHODS
The reporting of this paper follows the Strengthening the Reporting of OBservational studies in Epidemiology guideline.[31]

### Data source
This study used data from the Global School-based Student Health Survey (GSHS), developed by WHO and the Center for Disease Control and Prevention (CDC) in collaboration with UNICEF, UNESCO and UNAIDS.[32] The GSHS is a school-based survey conducted mostly among students aged 13–17 years, but some countries also include those aged 11–12 and 18 years.[33] Detailed information about the GSHS can be found on WHO and CDC websites.[34 35]

GSHS used a two-staged cluster sampling design to obtain a nationally representative sample of the adolescents. In the first stage, schools were randomly selected using the probability proportional to size sampling. In the second stage, classes were randomly selected in the selected schools, with varied number of classes depending on the school size. All students in the selected classes were eligible to participate in the survey, and the participation was anonymous and voluntary.

The GSHS surveys have been conducted in 185 mostly LMICs globally from 2003 to 2017. The survey consists of 10 core modules on adolescent health behaviours including physical activity and nutrition and protective factors, along with other optional modules. Data on response rates from each survey and characteristics of non-respondents were not available. Data cleaning and management, or data edits, were performed on all GSHS datasets: out of range edits, multiple response edits, logical consistency edits, height and weight edits, variable edits and record-level edits, where when the responses did not meet the requirement, they were set to missing. Observations with missing data were kept in the datasets. All GSHS datasets are freely available on WHO's website: https://wwwwhoint/teams/noncommunicable-diseases/surveillance/systems-tools/global-school-based-student-health-survey.

### Study design and ethics
This cross-sectional study used GSHS data from 31 Asian countries, as defined by the United Nations' Statistics Division of geographic regions.[36] All 31 countries assessed ATS using GSHS surveys.

### Data management
Data management was done using RStudio V.1.4. Prior to WHO data publication, out of range edits, multiple

response edits, logical consistency edits, height and weight edits, variable edits and record-level edits were performed on all GSHS datasets.[33] All 31 country datasets were drawn from the most recent GSHS survey in each country, except for Maldives and Oman which did not have national datasets for physical activity components in their most recent datasets. All 31 datasets were checked for missing values and observations with missing values were dropped.

## Study variables

The outcome variable, active travel to school, was self-reported using the question "During the past 7 days, on how many days did you walk or ride a bicycle to or from school?" with eight standard GSHS responses ranging from 0 to 7 days. In this study, travel to school was dichotomised into *passive* (0 days) and *active* (1–7 days) travel. This categorisation was chosen based on the distribution of ATS across the dataset (online supplemental figure S1), which showed high counts of 0 days, small counts of 1–4 days and higher counts of 5–7 days.

Independent variables included age, sex and BMI category. Age (in years) and sex (male or female) were self-reported. For analysis, age was categorised into two groups: *younger* (under 16 years old) and *older* (16 years old and over) adolescents. This cut-off point was as per standard GSHS reporting.[37] Sex was expressed as '1' for *male*, and '2' for *female*. Participants' height and weight were measured by survey staff before survey administration. Measurements were written by staff on slips of paper and given to each participant to be entered into their GSHS answer sheet. BMI category was generated based on WHO categorisation of BMI-for-age among children aged 5–19s, and then categorised into: *underweight and normal BMI* or *overweight and obesity*.[38]

## Statistical analysis

All statistical analyses were conducted using RStudio V.1.4. Only participants with complete data were included in the analyses (ranging from 34.2% to 97.4% by country). The 31 countries were categorised into three subregions: Eastern Mediterranean (which mostly consists of Middle Eastern countries); South East Asia (which consists of countries in the South and some countries in the Southeast Asia region) and Western Pacific (which consists of the rest of the countries in the Southeast Asia region along with countries in East and Central Asia). This categorisation was based on cultural and climate similarities shared by these countries, as well as WHO regional offices.[39] To adjust for non-response and distribution of adolescents by cluster, a sampling (survey) weight was applied to each adolescent. These weights were generated by accounting the size of cluster where the adolescent was sampled. Percentages and association estimates were also calculated under these weights to be representative of all students in each cluster. Logistic regression modelling was used to estimate each country's weighted associations of ATS, as independent variable, with age

group, sex and BMI category, as dependent variables. Younger (<16 years old) adolescents, male adolescents and adolescents with underweight or normal BMI category were used as the reference category. The weighted percentages and estimates (ie, OR) were pooled using random-effects meta-analysis to obtain the overall prevalence and estimates for the 31 countries. $I^2$ was used to determine the importance of heterogeneity in each meta-analysis (0%–40%: might not be important, 30%–60%: may represent moderate heterogeneity, 50%–90%: may represent substantial heterogeneity, 75%–100%: considerable heterogeneity).[40] We stratified the meta-analysis by subregion to establish potential regional differences. We also performed meta-regression to check the potential impact of year of survey variability. Year of survey, and sample size are shown on online supplemental table S1.

## Patient and public involvement

Patients or the public were not involved in the design, or conduct, or reporting, or dissemination plans of our research.

## RESULTS

Of the original 174 449 adolescents surveyed from the 31 countries, there were 152 368 (87.3%) adolescents aged 11–18 years with complete data included in the analysis. Participants were excluded due to missing values in one or more variables: n=893 for age, n=1518 for sex, n=19 996 for weight and height and n=2866 for travel to school. There was no significant difference between those included in this analysis, those excluded from this study and those excluded based on weight and height variables (online supplemental table S2). Table 1 shows the characteristics of adolescents included in the analyses, stratified by country. The proportion of younger adolescents varied across countries: from 50% in Thailand to 15% in Lao People's Democratic Republic. Across all countries, percentages of male participants were relatively equal, ranging between 45% and 50%. With respect to BMI category, Kuwait had the highest percentage of adolescents with overweight and obesity (51%), and Vietnam (6.2%) and Timor-Leste (6.2%) had the lowest.

## Overall active travel to school prevalence

The overall prevalence of adolescent ATS in Asia was 55% (see forest plot in online supplemental figure S2). Figures 1 and 2 show a map and a forest plot of the weighted prevalence of ATS in the 31 Asian countries, respectively. The forest plot is stratified by subregion and sorted from highest to lowest prevalence. In the Eastern Mediterranean, the ATS prevalence ranged from 18% (UAE) to 74.0% (Afghanistan), with a mean value of 47%. In the South East Asia, the prevalence of ATS ranged from 39% (Timor-Leste) to 84% (Myanmar), with a mean value of 56%. In the Western Pacific, ATS prevalence ranged from 25.0% (Brunei Darussalam) to 82.0% (China), with a mean value of 64%.

**Table 1** Characteristic of adolescents included in analyses from all 31 Asian countries

| Country | Population* | Age category† | | Sex | | BMI category‡ | |
|---|---|---|---|---|---|---|---|
| | | Younger | Older | Male | Female | Underweight and normal | Overweight and obesity |
| Eastern Mediterranean | | | | | | | |
| Afghanistan | 320973 | 34% | 66% | 54% | 46% | 83% | 17% |
| Bahrain | 67431 | 54% | 46% | 50% | 50% | 58% | 42% |
| Iraq | 1477045 | 51% | 49% | 57% | 43% | 73% | 27% |
| Jordan | 195315 | 37% | 63% | 52% | 48% | 76% | 24% |
| Kuwait | 152925 | 36% | 64% | 51% | 49% | 49% | 51% |
| Lebanon | 231936 | 44% | 56% | 46% | 54% | 72% | 28% |
| Occupied Palestinian territory | 255899 | 70% | 30% | 49% | 51% | 74% | 26% |
| Oman | 47730 | 30% | 70% | 47% | 53% | 76% | 24% |
| Pakistan | 2312737 | 61% | 39% | 61% | 39% | 92% | 8.1% |
| Qatar | 7546 | 81% | 19% | 52% | 48% | 50% | 50% |
| Syrian Arab Republic | 1229410 | 75% | 25% | 51% | 49% | 73% | 27% |
| UAE | 211732 | 38% | 62% | 49% | 51% | 59% | 41% |
| Yemen | 584111 | 57% | 43% | 64% | 36% | 88% | 12% |
| South East Asia | | | | | | | |
| Bangladesh | 4534799 | 64% | 36% | 64% | 36% | 89% | 11% |
| Bhutan | 59693 | 29% | 71% | 48% | 52% | 87% | 13% |
| India | 1529631 | 65% | 35% | 58% | 42% | 88% | 12% |
| Indonesia | 11585370 | 67% | 33% | 48% | 52% | 82% | 18% |
| Maldives and Male | 12521 | 35% | 65% | 49% | 51% | 88% | 12% |
| Myanmar | 1460394 | 65% | 35% | 45% | 55% | 92% | 8.4% |
| Nepal | 1869854 | 54% | 46% | 48% | 52% | 92% | 8.0% |
| Sri Lanka | 675391 | 39% | 61% | 46% | 54% | 85% | 15% |
| Thailand | 3134435 | 50% | 50% | 45% | 55% | 81% | 19% |
| Timor-Leste | 69318 | 22% | 78% | 50% | 50% | 94% | 6.2% |
| Western Pacific | | | | | | | |
| Brunei Darussalam | 25645 | 49% | 51% | 49% | 51% | 63% | 37% |
| Cambodia | 814883 | 31% | 69% | 52% | 48% | 96% | 3.8% |
| China | 721037 | 73% | 27% | 51% | 49% | 82% | 18% |
| Lao People's Democratic Republic | 287091 | 15% | 85% | 53% | 47% | 88% | 12% |
| Malaysia | 2181675 | 41% | 59% | 50% | 50% | 76% | 24% |
| Mongolia | 244281 | 49% | 51% | 48% | 52% | 88% | 12% |
| The Philippines | 6021848 | 49% | 51% | 49% | 51% | 90% | 10% |
| Vietnam | 7431982 | 21% | 79% | 47% | 53% | 94% | 6.2% |

*Population represented by the sample.
†Age was categorised into younger (<16 years old) and older (≥16 years old).
‡BMI category was dichotomised by WHO categorisation of BMI-for-age among children aged 5–19 years.
BMI, body mass index.

## Active travel to school association with age, sex and BMI category

Overall, older adolescents were 8% more likely to actively travel to school compared with younger adolescents (OR: 1.08; 95% CI: 1.00 to 1.16) (figure 3A). This association was strongest in the Eastern Mediterranean countries (OR: 1.13; 95% CI: 1.04 to 1.23), whereas no association was observed in the other two Asian subregions. Figure 3B shows that female adolescents were 21% less likely to actively travel to school than male adolescents (OR: 0.79; 95% CI: 0.71 to 0.89). Stratified analyses showed that this association was strongest in Eastern Mediterranean countries (OR: 0.67; 95% CI: 0.55 to 0.81), and then in South-East Asia countries (OR: 0.79; 95% CI: 0.69 to 0.90). No

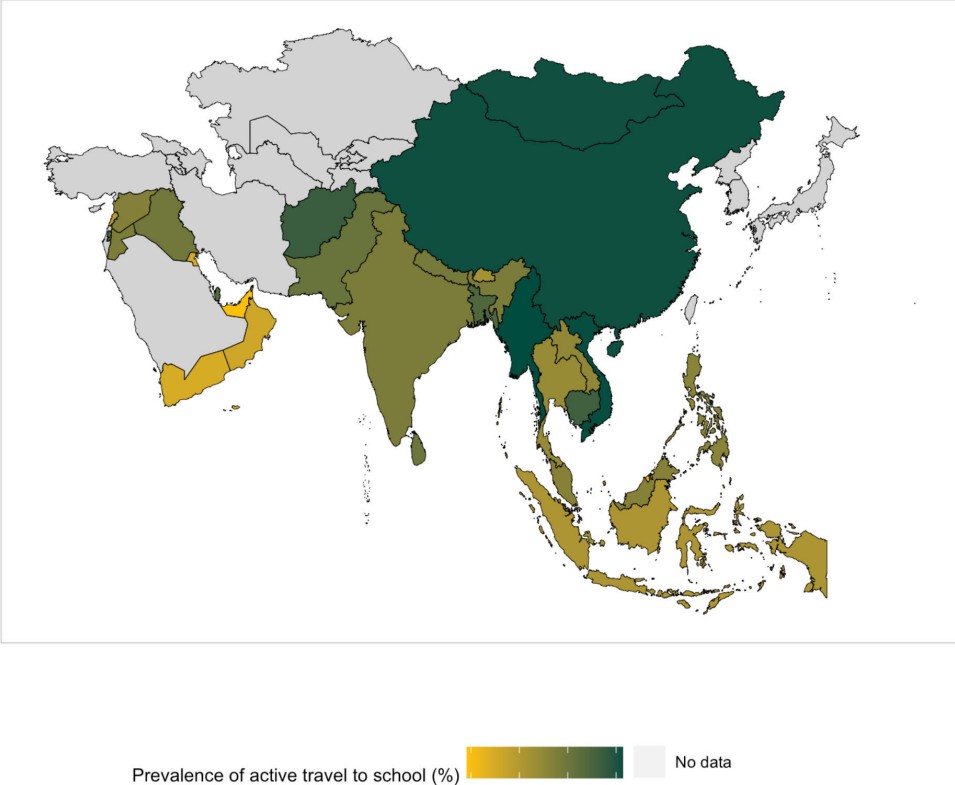

**Figure 1** Weighted prevalence of active travel to school in adolescents from 31 Asian countries using data from Global School-based Student Health Survey 2003–2017.

association was observed in Western Pacific countries. Lastly, adolescents with overweight and obesity were 8% less likely to actively travel to school than those with underweight or normal (OR: 0.92; 95% CI: 0.86 to 0.99) (figure 3C) weight categories. However, stratified analyses did not show any statistically significant associations by subregion.

### Heterogeneity

Overall heterogeneity ($I^2$) was considerable in all three meta-analyses (age group: 64%, sex: 87% and BMI category: 51%) (figure 3). Some region-specific variations were observed, with only the association with sex showing considerable heterogeneity (83%) in the Eastern Mediterranean, whereas considerable heterogeneity was observed across all meta-analyses in the Western Pacific. Meta-regression shows that year of survey accounted for some of this heterogeneity (online supplemental table S3).

### DISCUSSION
### Main findings

This study showed that 55% of Asian adolescents walk or cycle to school at least once per week. This prevalence varied substantially by country, with estimates varying from 18% (UAE) to 84% (Myanmar). However, regional variation was limited with 47% ATS prevalence noted in the Eastern Mediterranean, 56% in the South East Asia and

64% in the Western Pacific. Overall, older adolescents, male adolescents and adolescents with underweight/ normal weight were more likely to use active modes than their counterparts. However, these associations varied across countries, suggesting that there may be substantial country-level variation in determinants of ATS.

### Prevalence of active travel to school

The prevalence observed in the current study (55%) is broadly similar to that reported in previous studies from various countries and regions.[16 19 20] However, this is lower than global estimates in a recent review, which reported that 62% of boys and 54% girls actively travel to school.[41] Due to methodological differences between studies, our study's prevalence can only be compared directly with prevalence obtained from other GSHS studies[16 20] and comparison with other studies should be done with caution.

Despite limited regional variation, the Eastern Mediterranean had the lowest prevalence of adolescent ATS, which might reflect the high prevalence of physical inactivity in the general adolescent population in Eastern Mediterranean countries.[10] High physical inactivity, which includes ATS, may be due to the climate in the region. Evidence from research on adults in Saudi Arabia showed that half of participants were hesitant to walk daily due to the hot weather during the summer.[42] However, this evidence did not show that the hesitancy changed with

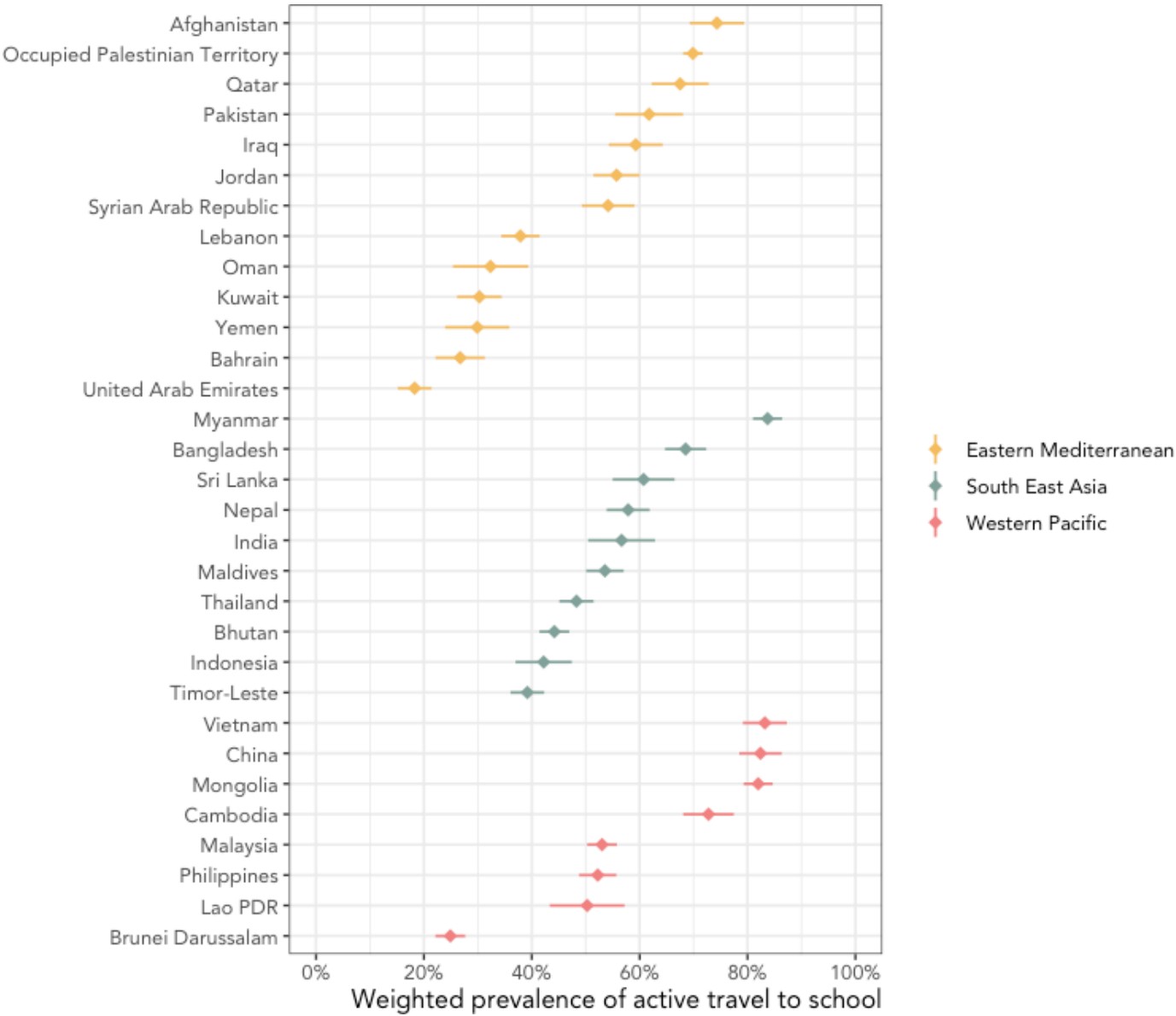

**Figure 2** Weighted prevalence of active travel to school across 31 Asian countries using data from Global School-based Student Health Survey 2003–2017. PDR, People's Democratic Republic.

changing seasons. Furthermore, other studies of children and adolescents conducted in European countries have shown that temperature, precipitation and wind speed were associated with various domains of physical activity, including ATS.[43–45]

### Differences by age, sex and BMI category

Similar to our findings, a study in Hong Kong reported that older adolescents were more likely to actively travel to school.[30] However, studies in other Asian countries have not reported differences in active travel between age groups.[26–29] Coincidentally, most of these previous studies come from Asian countries belonging to the Western Pacific, where we also found no difference of ATS prevalence between younger and older adolescents. In our study, difference in ATS prevalence by age group was also not noted in South East Asia countries but only present

in Eastern Mediterranean countries. A plausible explanation could be that the age when adolescents are permitted to travel independently differ by countries, perhaps due to varying perceptions of risks of injury or violence. For instance, age for independent mobility in South East Asia and Western Pacific countries may be lower than our study's age categorisation cut-off point. However, to our knowledge, no studies have explored this for the Asian region.

Findings from both our study and previous research using the GSHS and other global datasets demonstrated that female adolescents in Asia are less likely to actively travel to school than male adolescents.[25 27 46 47] Similarly, among US and European adolescents, female adolescents accumulated lower overall physical activity compared with male adolescents.[41] In general, female adolescents

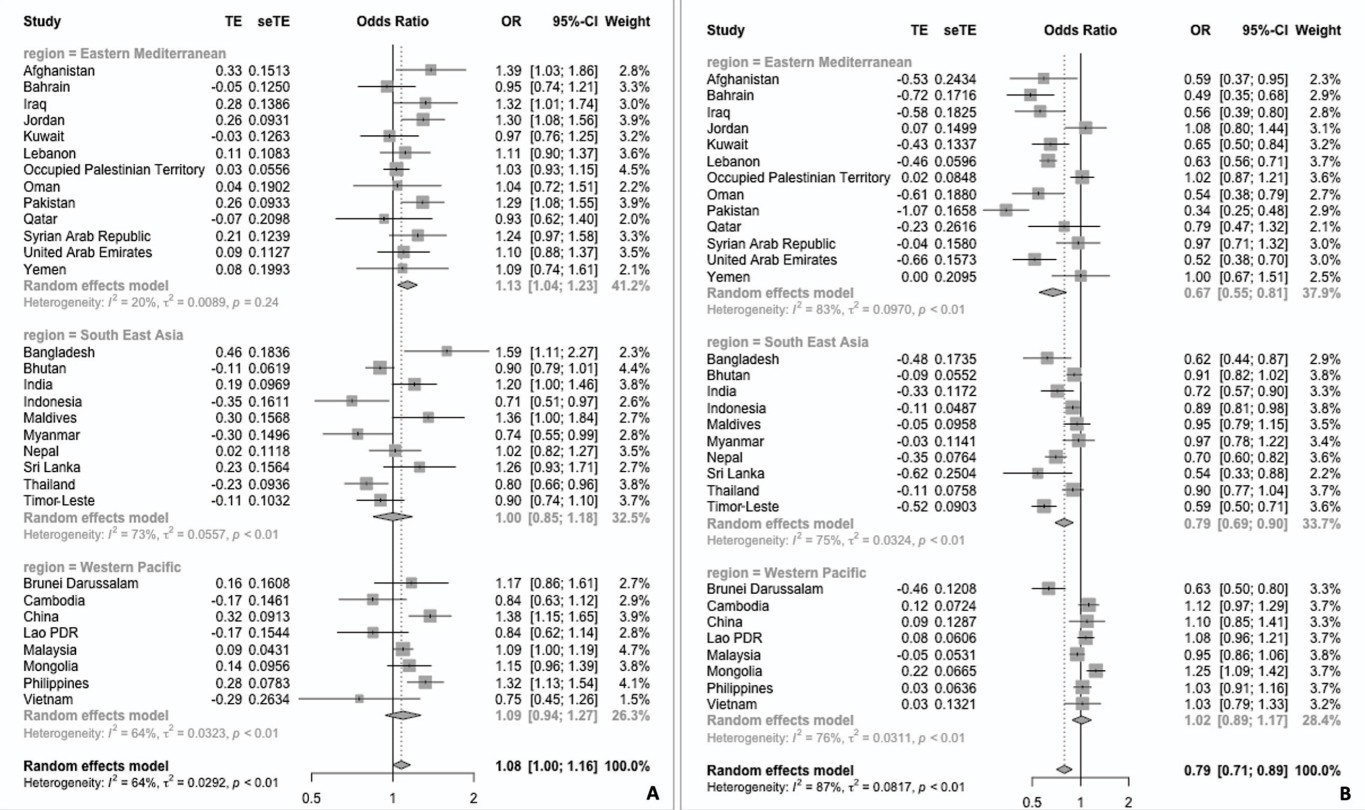

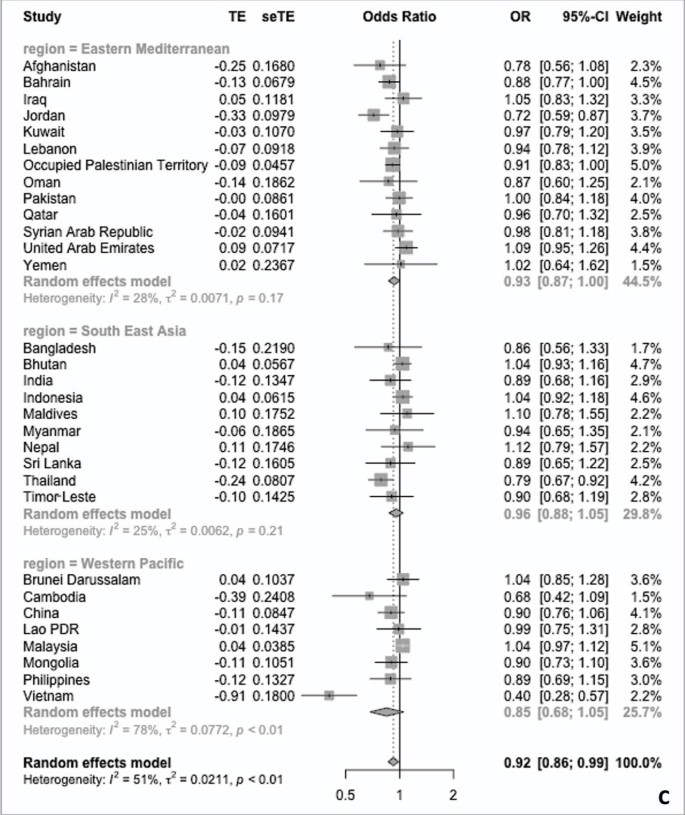

**Figure 3** Meta-analyses of logistic regressions for the associations of active travel to school and age, with younger adolescent as reference group (A), sex, with male as the reference group (B) and BMI category, with underweight and normal BMI as the reference group (C) in 31 Asian countries using data from Global School-based Student Health Survey 2003–2017. BMI, body mass index.

in Asia are more restricted in active travel due to cultural norms.[48] For example, a study among Indian adults showed that cycling is deemed appropriate for men but improper and unacceptable for women.[49] Another study from Saudi Arabia also reported that women's presence on the streets is limited because urban streets are considered a domain for men.[50] These cultural expectations may act as a barrier to increasing the prevalence of ATS among adolescent females in Asia.

Although adolescents with overweight and obesity BMI category were less likely to actively travel to school, this association was not observed in the subregion stratified analyses. Strong associations were only present in Cambodia and Vietnam, which is similar to findings from a Dutch population study where BMI was found to negatively influence later levels of active travel.[51] The results of our study suggest that BMI category may not be a significant determinant of ATS in most Asian countries. A plausible explanation for this finding could be that those adolescents with overweight and obesity BMI category are equally motivated to engage in active travel to obtain healthier BMI. However, the association of BMI and active travel may also be bi-directional. For example, a Danish adolescent study reported that cycling to school was associated with lower BMI.[52]

## Strengths and limitations

This study pooled estimates from 152 368 adolescents from 31 countries in Asia. Subregion and age-specific, sex-specific and BMI-specific estimates were also analysed. Data were collected using standardised sampling and data collection methods, enabling comparison across countries and subregions. However, the prevalence of ATS was obtained from one single self-reported question on both walking and cycling to school. Participating countries in the GSHS were also mostly those categorised as LMICs in Asia. As a result, the extent to which these findings are generalisable to other, particularly high-income, countries in Asia is unclear. This study also dichotomised both dependent and independent variables for meta-analysis purposes, which may have contributed to biases in the results, such as loss of information about individual differences and loss of effect.[53] Data on response rates from each survey and characteristics of non-respondents were not available, but we used the study weights provided to obtain representative estimates. Data collection was conducted in schools, and thus the conclusions drawn here are only relevant to adolescents in school, however out-of-school adolescents account for approximately 7% in Asia as compared with global percentage of 15%.[54] The GSHS also mainly focuses on adolescents aged 13–17 years, and the proportion of those aged 11–12 years was much smaller. Therefore, the evidence from adolescents in this younger age group may be under-represented. Lastly, the GSHS dataset does not contain information on individual-level socioeconomic position, which may be an important determinant of ATS.

## Implications

This evidence shows that there is substantial scope for improving ATS prevalence in Asian countries, which less than half of adolescents using ATS in many countries. The prevalence of ATS varied between and within the three Asian subregions, as did the associations with individual factors (ie, age group, sex and BMI category). This suggests that targeted promotion to certain population subgroups may be useful. However, future research is needed to explore why these variations occur among these countries to inform future policies and practices on ATS. Noting the variation of ATS by age group among the countries, age of independent mobility, especially on Eastern Mediterranean countries, should be studied so that future interventions with focus of age groups are appropriately designed. Future interventions should also focus more on designing ATS interventions that target female adolescents in Eastern Mediterranean and South East Asia countries. Any policy actions or interventions will need to be contextually sensitive, cognisant of local built, social and natural environmental contexts that could influence ATS. Further studies are therefore needed to gather evidence on the roles that environmental factors, such as exposure to air pollution, high-density traffic, walkability and safety, play in influencing ATS across Asian countries.[55] Such studies could build on child-friendly city initiatives[56] to generate evidence to inform strategies to equitably improve adolescent ATS across the region. Furthermore, there is a need to explore gendered cultural norms in the context of Asian countries, to ensure interventions designed to encourage ATS do not exclude female adolescents.

## Conclusions

The overall prevalence of adolescent ATS in Asia was 55%. The prevalence was lowest in UAE and highest in Myanmar. Overall, older adolescents, male adolescents and adolescents with underweight and normal BMI were more likely to actively travel to school than their counterparts. Although age, sex and BMI status were associated with the prevalence of ATS to varying degrees in the Eastern Mediterranean, South East Asia and Western Pacific regions, the main driver of variation within and between Asian countries remains unknown. Further investigations to identify other potential factors which account for differences in adolescent ATS prevalence across Asian countries is therefore needed to inform policy and practice.

**Acknowledgements** The authors would like to thank Stephen Sharp of MRC Epidemiology Unit, University of Cambridge and Ihsan Fadilah of Eijkman-Oxford Clinical Research Unit, Jakarta, for the support on data analysis, and Bimandra Djaafara of Imperial College, London, for the assistance on map creation.

**Contributors** RM designed the study with the supervision of EMFvS and TO. RM collected and managed the data assisted by EI. RM conducted the statistical analysis with guidance from EMFvS and EI. RM drafted the paper. All authors reviewed the results, edited the manuscript and agreed on the final version of the manuscript. RM is the guarantor of the study.

**Funding** RM's doctoral study is supported by Indonesia Endowment Fund for Education (201908220815174). EvS and EI are supported by the Medical Research Council (grant number MC_UU_00006/5). TO is supported by the National Institute for Health Research (NIHR) (16/137/34) using UK aid from the UK Government to support global health research.

**Disclaimer** The views expressed in this publication are those of the authors and not necessarily those of the funders. The funders had no role in the study.

**Map disclaimer** The inclusion of any map (including the depiction of any boundaries therein), or of any geographic or locational reference, does not imply the expression of any opinion whatsoever on the part of BMJ concerning the legal status of any country, territory, jurisdiction or area or of its authorities. Any such expression remains solely that of the relevant source and is not endorsed by BMJ. Maps are provided without any warranty of any kind, either express or implied.

**Competing interests** None declared.

**Patient and public involvement** Patients and/or the public were not involved in the design, or conduct, or reporting, or dissemination plans of this research.

**Patient consent for publication** Not applicable.

**Ethics approval** In each individual country, the GSHS was approved by the Ministry of Education, Ministry of Health or other institution in charge of the survey. Ethical approval was also obtained in all 31 individual countries. No reference numbers for these ethical approvals are available. Only those adolescents and their parents who provided written or verbal consent participated. As the current study used retrospective, de-identified, publicly available data, no separate ethics approval was required for the analysis of the data.

**Provenance and peer review** Not commissioned; externally peer reviewed.

**Data availability statement** Data are available in a public, open access repository. Data are available on reasonable request. Original GSHS datasets are available publicly online at https://www.who.int/teams/noncommunicable-diseases/surveillance/systems-tools/global-school-based-student-health-survey. Processed datasets used for this study and R codes used for data management and data analysis are available by emailing RM at rizka.maulida@mrc-epid.cam.ac.uk.

**ORCID iDs**
Rizka Maulida http://orcid.org/0000-0001-6964-8831
Erika Ikeda http://orcid.org/0000-0001-6999-3918

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
