## [Reviewer comments · BMJ Open]

ARTICLE DETAILS

TITLE (PROVISIONAL)	Descriptive epidemiology of the prevalence of adolescent active travel to school in Asia: a cross-sectional study from 31 countries
AUTHORS	Maulida, Rizka; Ikeda, Erika; Oni, T; van Sluijs, Esther

VERSION 1 – REVIEW

REVIEWER	Silvia Alejandra Gonzalez Cifuentes Universidad de los Andes Facultad de Medicina
REVIEW RETURNED	05-Nov-2021

GENERAL COMMENTS	Thanks for the opportunity to review this interesting and relevant manuscript. This study addressed the lack of Asian-specific estimates of active travel among school-attending adolescents. It provides a good addition to evidence by a rigorous estimation of active travel prevalence for Asian adolescents. I recommend including some minor revisions detailed below to improve the current manuscript. Please find a few comments and my suggestions below: Introduction The introduction provides a good background on the global situation of NCDS and physical inactivity, as well as the importance of promoting and maintaining active travel behaviours since adolescence. As a reader I appreciate the background provided on key characteristics of the continent that highlight the importance of focusing on this region. Page 5, lines 11-12, if words limit allow, I recommend expanding the idea of the dynamically changing built environments, or providing an example of what do the authors mean by this. Page 5, line 51, is it weight category or body mass index category? Methods The methods described the GSHS study and the main variables in a clear way. Looking at the description of the anthropometric variables and the categorization done, it seems more appropriate to refer to body mass index categories instead of only weight. But I may be misunderstanding. If this is not the case please provide more details in the description to avoid confusion. Page 9, line 9, I think there is an extra “the” before where the adolescent... Results Page 10, line 11. Was there any significant difference on key variables between the included and excluded participants? Table 1. If you are presenting BMI instead of only weight, I suggest using the category overweight and obesity instead of “above normal” Figure 1. Nice and useful map, I recommend to improve the title to make it more informative (e.g. include the source of data and maybe the time period)
--

	Figure 2. Again, good figure but I couldn't find the caption or title for it. Please provide a detailed caption. Also, please complete the legend for the X axis. Figure 3. It was difficult to read the information in light gray. Again, the captions were missing in the PDF received for review. Discussion The authors did a good summary of the main results and a very interesting interpretation of the results. However, I recommend deepening on the discussion of implications based on the associations found in this study. -Do you think that the lack of other relevant destinations for active travel in this age-group would be an important limitation in this study?
--	--

REVIEWER	Joanna Mazur University of Zielona Gora, Collegium Medicum
REVIEW RETURNED	09-Nov-2021

GENERAL COMMENTS	The paper describes a very interesting empirical material from 31 Asian countries, which are further divided into regions. The topic of ATS itself is now often analyzed and is closely related to the promotion of daily physical activity. I have the following comments:  1) Is it possible to talk about descriptive analysis, since there are elements of modeling (regression). Maybe changing „descriptive epidemiology” into "prevalence" in the title is reasonable. 2) The paper also has the advantage of sticking to STROBE Statement rigor. 3) Why the range of days of ATS is 0-7 days. Is it customary in all 31 countries for children to go to school on the weekend (especially Sunday) as well. 4) In describing limitations, attention was drawn to the dichotomous division of ATS, age and weight. I think it would be useful to also show (on a graph) the distribution of the number of days. Only one day of ATS is not indicative of favorable behavior, not indicative of an acquired habit. This is worth adding to the limitations or pointing out as a direction for further research. 5) Also, walking to school and bicycling are analyzed together, which is another limitation. In many papers, the authors focus on bicycle commuting. 6) It would also be useful to add the perspective of other non-Asian countries to the introduction, note that it is useful to compare Asian countries with high-income countries. However, a significant amount of the cited publications are from North America and Europe, including a splendid review [16]. 7) It seems to me that the discussion of differences between the 31 countries lacks depth. There is no basis from the data to point out the reasons. However, it was worth refining this part of the discussion. 8) I could find all titles of the figures and check them.
---

REVIEWER	Alfgeir Kristjansson West Virginia University, Department of Social and Behavioral Sciences
REVIEW RETURNED	09-Nov-2021

GENERAL COMMENTS	BMJ Open review This is an interesting observational study using secondary data from the GSHS survey of youth in 31 Asian countries. I found the study
--

	report mostly clear and to the point. My comments follow the order of sections in the manuscript. Abstract: Good and to the point. "ATS" should be spelled in full in the first line. Introduction: P4.Paragraph 1: After "developent" add "often" to state "...but their development often starts in childhood and..." P4.Paragraph 2: After "..activity". Add references to support claim. P4-P5L: Given that ATS analyses have already been conducted in multiple multi-country studies, clarification of what this study adds compared to the previous study reports would be helpful. P5.Paragraph 2: Paragraph lacks contextualization and a needs clear story line. P5L20-22: "...outcomes and risks associated with physical activity outdoors (24)". Please include examples. P5, paragraph 3: "Existing evidence....": But is that by design to to limited options? This reads like a reverse argument for your study criteria. P5, paragraph 3: "...favorable built environments..": Are the authors suggesting that poorer kids have "more favorable" built environments? Again, reverse argument. End of intro: Again, please clarify what this study adds to already existing studies/analyses into ATS in Asia. Methods Data source: Needs info on response rates between countries, attrition, and how missing data was dealt with. P7, paragraph 1: "..most recent GSHS survey..": Does this imply that the data is from multiple different time periods? If so, how is this dealt with in the analyses? P7, paragraph 2: Coding the outcome variable 0/1 with 0=0 and 1+=1 appears like a missed opportunity to do more refined analyses. Unclear why this analytical approach was selected and not multinomial logistic regression as one example. The categorization is not well justified despite the author's attempts. P7, bottom: "Only participants with complete data were included in the analyses". So how many were missing across different country data sets? At minimum the authors should present a range for these. P9, paragraph 1: Disproportionately large missing n for BMI measures. Discussing this as a limitation and justifying why this is not problematic for the reporting would be important. P9, paragraph 1: Delete "only" before "15%" (opinionated). Concluding paragraph Highlighting a need for "contextually tailored interventions" while also underlining that this study failed to find the "main driver" of
--	---

variation between the Asian countries" appears contradictory.

VERSION 1 – AUTHOR RESPONSE

Comments from Dr. Silvia Alejandra Gonzalez Cifuentes (Universidad de los Andes Facultad de Medicina), Reviewer 1:

No.	Comment	Response	Location
	Introduction		
3.	Page 5, lines 11-12, if words limit allow, I recommend expanding the idea of the dynamically changing built environments, or providing an example of what do the authors mean by this.	We have revised this sentence as suggested	Page 5
4.	Page 5, line 51, is it weight category or body mass index category?	We have discussed your suggestion and decided that BMI category is most appropriate. Thus, we have changed the “weight category” to “BMI category”.	Throughout the manuscript, tables, and figures
	Methods		
5.	Looking at the description of the anthropometric variables and the categorization done, it seems more appropriate to refer to body mass index categories instead of only weight. But I may be misunderstanding. If this is not the case please provide more details in the description to avoid confusion.	We have made this change, please also see response to Comment No. 4	-
6.	Page 9, line 9, I think there is an extra “the” before where the adolescent...	We have revised this sentence.	Page 8, middle of paragraph
	Results		
7.	Page 10, line 11. Was there any significant difference on key variables between the included and excluded participants?	We have added Supplementary Table S2 in the Supplementary File comparing age group, sex, and ATS of adolescents included in the study, excluded from the study, and in the original dataset. This shows that there is no substantial difference. We did not compare BMI due to the significant amount of	Page 9, middle of first paragraph Supplementary Table S2

		missing value in this variable. We have added this to the Results section.	
8.	Table 1. If you are presenting BMI instead of only weight, I suggest using the category overweight and obesity instead of “above normal”	We have revised “above normal” to “overweight and obesity”.	Throughout the manuscript, tables, and figures
9.	Figure 1. Nice and useful map, I recommend to improve the title to make it more informative (e.g. include the source of data and maybe the time period)	The caption now reads “Weighted prevalence of active travel to school in adolescents from 31 Asian countries using data from Global School-based Student Health Survey 2003 to 2017”. For clarity, we now also list all figure titles before the References section.	Figure 1 and page 19
10.	Figure 2. Again, good figure but I couldn’t find the caption or title for it. Please provide a detailed caption. Also, please complete the legend for the X axis.	We have added titles of the figures before the references. We also have revised the legend for X axis.	Page 19 Figure 2
11.	Figure 3. It was difficult to read the information in light gray. Again, the captions were missing in the PDF received for review.	We have improved the readability of Figure 3. We have added captions of the figures before the References section.	Figure 3 and page 19
	Discussion		
12.	The authors did a good summary of the main results and a very interesting interpretation of the results. However, I recommend deepening on the discussion of implications based on the associations found in this study.	Thank you for your positive feedback. We have expanded the Implications subsection of the Discussion section.	Page 14
13.	Do you think that the lack of other relevant destinations for active travel in this age-group would be an important limitation in this study?	Our study focused on active travel to school thus the lack of other destinations for active travel in the GSHS was not a limitation. Additionally, as stated in Gonzalez et al’ study [19], travel to and from school is the main trips for children and adolescents.	-

Comments from Dr. Joanna Mazur (University of Zielona Gora), Reviewer 2:

No.	Comments	Responses	Location
14.	Is it possible to talk about descriptive analysis, since there are elements of modeling (regression). Maybe changing „descriptive epidemiology” into "prevalence" in the title is reasonable.	We have amended the title and the aim to “Descriptive epidemiology of the prevalence of adolescent active travel to school in Asia: a cross-sectional study from 31 countries”	Page 1 (title) and page 5 (aim)
15.	The paper also has the advantage of sticking to STROBE Statement rigor.	Thank you very much for this comment.	-
16.	Why the range of days of ATS is 0-7 days. Is it customary in all 31 countries for children to go to school on the weekend (especially Sunday) as well.	The answer categories for the ATS variable in the GSHS dataset is 0-7 days for all countries, not just these 31 countries. Schools in some countries also have extracurricular activities on weekends thus the 0-7 days range is inclusive. For clarity, we have included “eight standard GSHS responses” to the Study variables subsection in Methods section.	Page 7, third paragraph
17.	In describing limitations, attention was drawn to the dichotomous division of ATS, age and weight. I think it would be useful to also show (on a graph) the distribution of the number of days. Only one day of ATS is not indicative of favorable behavior, not indicative of an acquired habit. This is worth adding to the limitations or pointing out as a direction for further research.	We now present the distribution of ATS in the supplementary file (Figure S1). While one day of active travel is not indicative of an acquired habit, we decided to include it as active travel rather than passive due to the distribution of the active travel data which shows a bimodal distribution. We initially categorized ATS into 3 groups (0 = passive, 1-4 = occasional, 5-7 = frequent). Due to the complexity of the sampling frame, it was not possible to analyze the data all together thus we ran country-level analyses. Using three outcomes created challenges for subsequent. Therefore, we decided to use 2 outcomes instead (0 vs 1-7).	Supplementary file (Figure S1)
18.	Also, walking to school and bicycling are analyzed together, which is another limitation. In many papers, the authors focus on bicycle commuting.	Walking to school and bicycling to school were asked as one question in GSHS thus it was not possible to separate them. We have added this as a potential limitation to the Strengths and limitations section.	Page 13

19.	It would also be useful to add the perspective of other non-Asian countries to the introduction, note that it is useful to compare Asian countries with high-income countries. However, a significant amount of the cited publications are from North America and Europe, including a splendid review [16].	We have included a study from the 63 low- and middle-income countries (LMICs) in addition to the systematic review study in the Introduction. However, many studies cited in the introduction section are multi-countries study which included non-Asian LMICs in them.	Page 4, third paragraph
20.	It seems to me that the discussion of differences between the 31 countries lacks depth. There is no basis from the data to point out the reasons. However, it was worth refining this part of the discussion.	This study was set up to identify differences between and within sub-regions, however, as mentioned, the main driver of these differences is still unknown. We believe that in-depth discussion is beyond the scope and data presented here; the reasons raised in the discussions were extrapolation from previous studies.	-
21.	I could find all titles of the figures and check them.	For clarity, we now present the Figure titles before the References section.	Page 19

Comments from Dr. Algeir Kristjansson (West Virginia University, Icelandic Center for Social Research and Analysis), Reviewer 3:

No.	Comments	Responses	Location
	Abstract		
22.	Good and to the point. "ATS" should be spelled in full in the first line.	We have spelled out ATS in full. The change can be seen in the objective subsection of the abstract. Please note that the abstract has been revised to fit with the author instructions.	Page 2
	Introduction		
23.	P4.Paragraph 1: After "development" add "often" to state "...but their development often starts in childhood and..."	We have added "often" after "development". The change can be seen at the bottom of the first paragraph of the Introduction section.	Page 4
24.	P4.Paragraph 2: After "..activity". Add references to support claim.	This sentence referred to references 14 and 15. We have moved references 14 and 15 to the end of the sentence.	Page 4
25.	P4-P5L: Given that ATS analyses have already been conducted in multiple multi-country studies,	We recognize that other analyses have been published. However, as stated in third paragraph in the	Page 4-5

	clarification of what this study adds compared to the previous study reports would be helpful.	Introduction section (page 4), Gonzalez et al' study [19] was unable to generate a pooled estimate of the global prevalence of active travel due to different methodologies used in the countries studied. Moreover, as stated in the third paragraph of the Introduction section, Peralta et al' study [16] generated overall prevalence of the 63 low- and middle-income countries from various regions, but the study did not have a specific overall prevalence for Asia as well as comparisons between countries. While Uddin et al [20] examined each country's ATS, countries included were of some South Asian countries, Southeast Asian countries, Mongolia, and Pacific countries. Furthermore, this study did not examine an overall prevalence of the countries in Asia, as well as comparison within the continent. Our study generated an overall prevalence of Asia, as well as sub-regional and individual country prevalence. In addition, the previous studies did not examine ATS prevalence differences by age, sex, and BMI category between and within the Asian subregions. We have revised the third paragraph of the Introduction section for clarity.	
126.	P5.Paragraph 2: Paragraph lacks contextualization and a needs clear story line.	We have revised this paragraph to improve the flow.	Page 5
27.	P5L20-22: "...outcomes and risks associated with physical activity outdoors (24)". Please include examples.	Cities in Asia have the worst air pollution levels thus outdoor physical activity can increase the risk of respiratory and cardiovascular diseases, as well as cancer. We have added examples and revised this sentence.	Page 5
28.	P5, paragraph 3: "Existing evidence....": But is that by design to to limited options? This reads like a reverse argument for your study	There was a few limited studies done in countries within Asia and these studies reported some associations. What we meant here was evidence from these limited Asian studies. We	Page 5

	criteria.	have revised this paragraph for clarity.	
29.	P5, paragraph 3: "...favorable built environments..": Are the authors suggesting that poorer kids have "more favorable" built environments? Again, reverse argument.	What we meant here was that both favourable built environments and students living in less wealthy areas or studying in less wealthy areas were reported to be associated with more active travel to school. However, these factors were associated separately. We have revised this paragraph for clarity.	Page 5
30.	End of intro: Again, please clarify what this study adds to already existing studies/analyses into ATS in Asia.	This has been revised, please see response to comment No. 25	-
	Methods		
31.	Data source: Needs info on response rates between countries, attrition, and how missing data was dealt with.	We have added the details to third paragraph of Data source subsection in Methods section. However, please note that the analyses are weighted and account for differential non-response.	Page 6
32.	P7, paragraph 1: "..most recent GSHS survey..": Does this imply that the data is from multiple different time periods? If so, how is this dealt with in the analyses?	Yes, data were taken from multiple time periods. We checked with meta-regression for the potential impact of year of survey variability (Supplementary Table S3). While year of survey accounted for some of this heterogeneity, it was not the main driver of variation. Therefore, no change was made.	-
33.	P7, paragraph 2: Coding the outcome variable 0/1 with 0=0 and 1+ =1 appears like a missed opportunity to do more refined analyses. Unclear why this analytical approach was selected and not multinomial logistic regression as one example. The categorization is not well justified despite the author's attempts.	We now present the distribution of ATS in the supplementary file (Figure S1). We initially categorized ATS into 3 groups (0 = passive, 1-4 = occasional, 5-7 = frequent). Due to the complexity of the sampling frame, it was not possible to analyze the data all together thus we ran country-level analyses. Using three outcomes (with multinomial logistic regression) created challenges in summarizing the estimates with meta-analysis. Therefore, we decided to use 2 outcomes (0 vs 1-7) and use logistic	Supplementary file (Figure S1)

		regression instead.	
34.	P7, bottom: "Only participants with complete data were included in the analyses". So how many were missing across different country data sets? At minimum the authors should present a range for these.	As suggested, we have added the range for non-response in the Statistical analysis subsection in Methods section.	Page 8
35.	P9, paragraph 1: Disproportionately large missing n for BMI measures. Discussing this as a limitation and justifying why this is not problematic for the reporting would be important.	We have added Supplementary Table S2 in the Supplementary File comparing age group, sex, and ATS of adolescents included in the study, excluded from the study, and in the original dataset. There is no significant difference when we compared the characteristics of adolescent missing weight and height and those included in the analysis. We also used the study weights provided to obtain representative estimates. Therefore, we believe this is not a limitation for our study.	Supplementary file (Table S2)
36.	P9, paragraph 1: Delete "only" before "15%" (opinionated).	We have deleted the word "only" before "15%". The change can be seen in the middle of the first paragraph of the Results section.	Page 9
37.	Concluding paragraph Highlighting a need for "contextually tailored interventions" while also underlining that this study failed to find the "main driver" of variation between the Asian countries" appears contradictory.	We have amended this paragraph to be more consistent.	Page 14

VERSION 2 – REVIEW

REVIEWER	Silvia Alejandra Gonzalez Cifuentes Universidad de los Andes Facultad de Medicina
REVIEW RETURNED	25-Jan-2022

GENERAL COMMENTS	I recommend the publication of this manuscript, it presents important findings, and the minor concerns highlighted in the first revision were properly addressed and solved. This paper will add relevant evidence to the active transportation literature.
---

REVIEWER	Joanna Mazur University of Zielona Gora, Collegium Medicum
REVIEW RETURNED	20-Jan-2022

GENERAL COMMENTS	The authors clearly addressed the comments from all 3 reviews ,
---

	including mine. This is evidenced by the table of responses and changes marked in the text. I found the paper to be of great value because of its geographical coverage.
REVIEWER	Alfgeir Kristjansson West Virginia University, Department of Social and Behavioral Sciences
REVIEW RETURNED	18-Jan-2022
GENERAL COMMENTS	The authors have responded adequately to my earlier comments.